# Hotspot siRNA Confers Plant Resistance against Viral Infection

**DOI:** 10.3390/biology11050714

**Published:** 2022-05-06

**Authors:** Atef Mohamed, Zhenhui Jin, Toba Osman, Nongnong Shi, Mahmut Tör, Stephen Jackson, Yiguo Hong

**Affiliations:** 1School of Life Sciences, University of Warwick, Coventry CV4 7AL, UK; toba.osman@yahoo.co.uk (T.O.); nshi63@hznu.edu.cn (N.S.); stephen.jackson@warwick.ac.uk (S.J.); 2Department of Botany, Faculty of Agriculture, Fayoum University, Fayoum 63514, Egypt; 3School of Life and Environmental Sciences, Hangzhou Normal University, Hangzhou 311121, China; jinz1_20@uni.worc.ac.uk; 4School of Science and the Environment, University of Worcester, Worcester WR2 6AJ, UK; m.tor@worc.ac.uk

**Keywords:** ACMV, siRNA, RNAi, RDR6, antiviral defence

## Abstract

**Simple Summary:**

A hallmark of antiviral RNAi is the production of viral siRNA (vsiRNA). Profiling of vsiRNAs indicates that certain hotspot regions of viral genome or transcribed viral RNAs are more prone to RNAi-mediated cleavage. However, the biological relevance of hotspot vsiRNAs to the host innate defence remains to be elucidated. Here, we show that direct targeting a hotspot by synthetic vsiRNA confers plant resistance to virus infection. Hotspot and coldspot vsiRNAs, based on vsiRNA profile of the African cassava mosaic virus (ACMV), were synthesised. However, only the double-stranded hotspot vsiRNA protected plants from ACMV infection with undetectable levels of viral DNA replication and viral mRNA. We further demonstrated that the hotspot vsiRNA-mediated virus resistance had a threshold effect and required an active *RDR6*. These data show that hotspot vsiRNAs bear a functional significance on antiviral RNAi, suggesting that they may have the potential as exogenous protection agents for controlling destructive plant viral diseases.

**Abstract:**

A hallmark of antiviral RNA interference (RNAi) is the production of viral small interfering RNA (vsiRNA). Profiling of vsiRNAs indicates that certain regions of viral RNA genome or transcribed viral RNA, dubbed vsiRNA hotspots, are more prone to RNAi-mediated cleavage for vsiRNA biogenesis. However, the biological relevance of hotspot vsiRNAs to the host innate defence against pathogens remains to be elucidated. Here, we show that direct targeting a hotspot by a synthetic vsiRNA confers host resistance to virus infection. Using Northern blotting and RNAseq, we obtained a profile of vsiRNAs of the African cassava mosaic virus (ACMV), a single-stranded DNA virus. Sense and anti-sense strands of small RNAs corresponding to a hotspot and a coldspot vsiRNA were synthesised. Co-inoculation of *Nicotiana benthamiana* with the double-stranded hotspot siRNA protected plants from ACMV infection, where viral DNA replication and accumulation of viral mRNA were undetectable. The sense or anti-sense strand of this hotspot vsiRNA, and the coldspot vsiRNA in both double-stranded and single-stranded formats possessed no activity in viral protection. We further demonstrated that the hotspot vsiRNA-mediated virus resistance had a threshold effect and required an active *RDR6*. These data show that hotspot vsiRNAs bear a functional significance on antiviral RNAi, suggesting that they may have the potential as an exogenous protection agent for controlling destructive viral diseases in plants.

## 1. Introduction

RNAi involves double-stranded (ds) or structured single-stranded (ss) RNA from which short interfering RNA (siRNA) duplexes are derived through the action of RNase III-like DICER or Dicer-like (DCL) enzymes [1,2,3]. Once unwounded, the guide strand of siRNA duplex is incorporated into an RNA-induced silencing complex (RISC), and then pairs with the target RNA, leading to its cleavage in a sequence-specific manner by a slicer Argonaute protein, one of the key RISC components [4]. In plants and invertebrates, RNAi plays an essential role in innate immune defence against viral pathogens. Viruses can trigger RNAi which subsequently targets and degrades viral RNA to produce primary and secondary virus-derived siRNA (vsiRNA) of 21 to 24 nucleotides (nt) in length [5]. Biogenesis of the primary and secondary vsiRNAs involves DCLs and may require host RNA-dependent RNA polymerases (RDRs), such as RDR6 [6]. There is also emerging evidence that biogenesis of vsiRNAs occurs in mammalian cells infected with a number of RNA viruses, representing a novel dimension in both viral pathogenesis and host antiviral response in vertebrates [7]. However, despite occurrence of vsiRNA hotspots is a generic phenomenon, little is known about the function of hotspot vsiRNAs in cellular defence against viral attack across the plant and animal kingdoms.

Africa cassava mosaic virus (ACMV) is a geminivirus in the genus *Begomovirus*, family *Geminiviridae*. The bipartite ACMV genome consists of two circular ssDNA components (DNA-A and DNA-B) that share an almost identical sequence of approx. 200 nucleotides containing the replication origin and regulatory elements essential for viral gene transcription, and are required for systemic infection [8,9]. ACMV causes cassava mosaic disease (CMD) and can result in massive economic losses in the production of cassava *(Manihot esculenta* Crantz), an essential staple carbohydrate food source in Africa. Tremendous efforts have been made to generate CMD-resistant cassava cultivars. However, conventional breeding programmes to control ACMV have been largely unsuccessful. Contemporary genetic engineering may provide a means to produce virus-resistant plants, but the public scepticism of the transgenic technology prevents its potential use. Therefore, there is a need for the development of novel alternative strategies to control CMD. ACMV can be the inducer and target of RNAi, and vsiRNAs are found to be associated with ACMV infection in cassava and *Nicotiana benthamiana* [10,11].

In this study, we profiled ACMV siRNAs and demonstrated the existence of vsiRNA hotspots in the ACMV genome. Direct targeting viral genome with a synthetic hotspot vsiRNA protected plants from ACMV infection. We further showed that the hotspot vsiRNA-mediated viral resistance possessed a threshold effect and required a functional *RDR6*.

## 2. Materials and Methods

### 2.1. Virus Infection and Plant Maintenance

Wild-type *N. benthamiana* and transgenic line NbRDR6i plants were mechanically inoculated with infectious clones of ACMV (pCLV1.3A and pCLV2B) [12] together with or without synthetic siRNAs. A total of 1.0 µg of ACMV (a mixture of purified plasmid DNA of 500 ng pCLV1.3A and 500 ng pCLV2B) alone or together with 0–20 µg sense strand, antisense strand, or double-stranded siRNA in 20 µL DNase/RNase-free TE (10 mM Tris-HCl, 1 mM EDTA, pH 8.0) were pipetted onto young leaves of plants at the 6-leaf stage, and gently rubbed by fingers against the leaf surface pre-dusted with a thin layer of fine carborundum powder. Plants were maintained in an insect-free containment glasshouse or growth room at 25 °C with supplementary lighting to give a 16 h day/8 h night photoperiod. Local and systemic symptom development was assessed on a daily basis and was photographically recorded with a Nikon Digital Camera Coolpix 995. 

### 2.2. Detection of ACMV siRNAs

Total RNAs were isolated from approx. 1 g of plant leaf tissues using TRIZOL reagent (Invitrogen, Waltham, MA, USA) according to the manufacturer’s instructions. Small RNAs including vsiRNAs were separated from high molecular weight RNAs and dissolved in 20 μL RNase-free water as described [1]. Small RNA samples (2 μg) were mixed with 2 × RNA loading buffer (New England Biolabs, Ipswich, MA, USA), denatured at 95 °C for 5 min and then chilled on ice for 5 min before resolved on a 15% polyacrylamide-8 M urea gel in TBE buffer (89 mM Tris-borate pH8.3, 20 mM EDTA) at 120 V for 2.5 h. Small RNAs were transferred onto Hybond N membrane (Amersham) by overnight capillary blotting, UV cross-linked, prehybridized in prehybridization buffer [50% formamide, 1% N-laurylsacosine, 7.6% SDS and 2% blocking reagent (Roche, Basel, Switzerland)] at 40 °C for 3 h, and then hybridized at 95 °C overnight with the digoxigenin (DIG)-labelled viral specific probes. Five different probes (A1-A5 and B1-B5) for ACMV DNA-A or DNA-B, respectively, were prepared by PCR using DIG-11-dUTP (Roche) and various sets of primers (Table 1) as previously described [13,14]. After the hybridization, membranes were washed in 2 × SSC/0.5% SDS for 15 min at 40 °C twice and then in 0.5 × SSC/0.5% SDS for 15 min at 40 °C twice. The hybridization signal was then detected using the DIG Luminescent Detection Kit (Roche) and exposed to Kodak X-ray films following the manufacturer’s instructions.

### 2.3. RNAseq Profiling of ACMC vsiRNAs

Total RNAs were extracted from ACMV infected *N. benthamiana* leaves at 21 days post-inoculation and fractionated as described [1]. The small RNA bands were gel-excised and ligated to a pair of 5′ and 3′ adapters. After the reverse transcription and amplification by one-step RT-PCR, DNA fragments were separated on a 3.5% agarose gel. The timer bands containing “454” 5′-primer/small RNA/”454” 3′-primer (approx. 125 base pairs) were excised, quantified and sequenced by “454 sequencer” and subsequent analysis of vsiRNAs were carried out as described [15]. Briefly, the clean sRNA sequences were blasted against both ACMV DNA A and DNA B [8], and specific vsiRNAs were plotted along the ACMV DNA A and DNA B genomes using the MiniTab^®^-14 programme [15].

### 2.4. In Vitro Production of siRNA

We designed T7 promoter-bearing olingonucleotides for synthesis of sense- and anti-sense strands of small RNAs with 3′ UU overhang that were selected to target a hotspot and a non-hotspot vsiRNA region of ACMV DNA-A (Figure 1). Double-stranded oligonucleotides were obtained though annealing an equal amount of primers in the following pairs PP768/PP769, PP770/771, PP772/PP773, PP774/PP775 (Table 1), respectively. Single-stranded small RNAs were produced by in vitro transcription in a mixture containing 1 × transcription buffer (New England Biolabs), 10 μg double-stranded oligonucleotide template, 2 mM of GTP, ATP, CTP and UTP, 200 units T7 RNA polymerase (New England Biolabs, Ipswich, MA, USA) and 40 units RNasin (Promega, Madison, WI, USA). The reaction was performed at 37 °C for 1 h, followed by treatment of RNase-free DNase (Promega) at 37 °C for 15 min. Small RNAs were purified by phenol/chloroform extraction and re-suspended in 20 μL of RNase-free water. Concentrations of the sense and antisense small RNA products were measured by Nanodrop and their integrities were analysed on 15% polyacrylamide-8M urea gel. An equal amount of sense and antisense small RNAs was annealed to form ds-siRNA.

### 2.5. DNA Extraction and Southern Blot

DNA was extracted from inoculated leaves and leaves systemically infected with ACMV at 14 days post-inoculation with a DNeasy Plant Mini Kit (Qiagen, Hilden, Germany). DNA aliquots (150 ng) were resolved on a 1.0% agarose gel in TAE buffer (40 mM Tris-acetate pH 7.5, 20 mM acetic acid, 1 mM EDTA), transferred to nylon membrane and detected by hybridization with DIG-labelled probes specific to either ACMV DNA-A or DNA-B with a DIG DNA labelling and detection kit (Roche) as described [16].

## 3. Results

### 3.1. Profiling ACMV vsiRNAs

In ACMV-infected *N. benthamiana*, two main size-classes of 21 and 24 nt vsiRNAs were constantly detectible by northern hybridizations using probes specific to both DNA-A (Figure 2A) and DNA-B (Figure 2B). VsiRNAs were found to be associated with the DNA-A and DNA-B components and they were readily detected in total small RNAs extracted from plants at 12 days post-inoculation (dpi) and accumulated to a relatively high level at 21 dpi. However, DNA-B originated vsiRNAs seemed to be less abundant compared to that corresponding to DNA-A. To further characterise ACMV vsiRNAs, we sequenced small RNAs extracted from *N. benthamiana* plants infected with ACMV using the “454” sequencing technology. In total, there were 28,257 vsiRNAs against DNA-A and 9452 vsiRNAs against DNA-B, consistent with the differential abundance of vsiRNAs of the two genome components detected by Northern blots (Figure 2A,B). However, the size of these vsiRNAs ranged from 15 to 29 nt, and was dominated by 17 to 20 nt. Nevertheless, the high-throughput sequencing analysis demonstrated that vsiRNAs were corresponding to genomic regions with or without protein coding capacity, and showed distribution in the sense and complementary sense strands of both genome components (Figure 2C,D). Moreover, vsiRNAs of various sizes could be derived from the same viral genome locations (Figure 2C,D), suggesting that different DCLs were involved in process of ACMV RNA transcripts for the biogenesis of these vsiRNAs [10,11]. Most significantly, the distribution of vsiRNAs showed that obvious hotspots, from which a large number of specific vsiRNAs were generated, existed in both ACMV DNA-A (Figure 2C) and DNA-B (Figure 2D). Interestingly, there were also regions, dubbed coldspots, within the viral genome to which no matching vsiRNA was found. These data suggest that specific sequences, and likely their surrounding structures of viral transcribed RNAs could be differentially targeted by the RNAi machinery for biogenesis of vsiRNAs.

### 3.2. Hotspot siRNA-Mediated Viral Resistance

The manifestation of vsiRNA hotspots in the viral genome raises a possibility that the host cellular RNAi machinery may be able to target and cleave RNA at these hotspots in order to counter viral attack. To test this idea, we selected a particular vsiRNA hotspot locating at nucleotides 1458 to 1440 of the complementary strand of DNA-A and an adjacent coldspot region from nucleotides 1359 to 1341 of the same strand, to which no vsiRNA was found, and designed an experimental strategy for in vitro synthesis of siRNAs with 3′ UU overhang (Figure 1 and Figure 3A). Sense and antisense strand of the two siRNAs, designated hotspot and coldspot siRNA respectively, were produced by in vitro transcription using T7-promoter controlled dsDNA templates. The authenticity of the synthetic hotspot and coldspot small RNAs were scrutinized by the treatment of DNase I and their sizes were checked by denaturing 8M urea-PAGE (Figure 3B). Ds-siRNA was obtained through annealing an equal amount of sense and antisense strands of small RNA transcripts.

To assess the impact of the hotspot and coldspot siRNAs on viral infection, we challenged *N. benthaniana* plants with ACMV alone or together with siRNAs. Compared with mock-inoculation (Figure 3C), ACMV was able to establish local infection, evident by the development of chlorotic lesions on the inoculated leaves at 5–7 dpi. Systemic symptoms including mosaic, leaf distortion and plant dwarfing appeared at 10–14 dpi (Figure 3D). Meanwhile we did not observe any impact of sense or antisense, hotspot or coldspot ss-siRNAs on ACMV infection, and in all cases, co-inoculated plants developed typical local and systemic disease phenotypes (Figure 3E–H). However, plants co-inoculated with ACMV and hotspot ds-siRNA remained symptomless (Figure 3I). On the other hand, the coldspot ds-siRNA was incapable of defending plants from ACMV infection and plants exhibited severe viral symptoms (Figure 3J). These results were reproducible (Table 2). The level of viral DNA in ACMV-infected plants was further investigated by Southern blot analysis of inoculated and young leaves sampled at 14 days after the virus challenge using DNA-A- and DNA-B-specific probes (Figure 3K–P). Consistent with symptom severity, accumulation of both viral genomic components was readily detectable in plants where sense or antisense strands of hotspot (Figure 3K–M) or coldspot siRNA, as well as coldspot ds-siRNA (Figure 3N–P) provided no protection from viral infection. On the other hand, these viral DNAs were not detected in asymptomatic tissues of plants co-inoculated with ACMV and hotspot ds-siRNA (Figure 3K–M). This is in contrast to randomly designed siRNA that could interfere with viral gene expression and viral DNA accumulation in cultured plant cells, but the efficiency of inhibiting viral replication was low [17]. Moreover, the residual of inoculum ACMV DNA in the inoculated leaves but not the newly growing young leaves (Figure 3K,L) suggested that simultaneously introducing the synthetic hotspot ds-siRNA activated the host RNAi defence mechanism that effectively repressed ACMV proliferation in inoculated leaves, and subsequently prevented ACMV from establishing a systemic infection in plants.

### 3.3. Threshold Effect of Hotspot ds-siRNA-Mediated ACMV Suppression

To further investigate the effect of hotspot ds-siRNA on viral infection, we challenged *N. benthaniana* plants with 1.0 µg of ACMV and 0, 2.5, 5.0, 10.0 or 20.0 µg of hotspot ds-siRNA, respectively, in three separate experiments. In contrast to mock-inoculated plants (Figure 4A), local and systemic symptoms developed in all plants co-inoculated with ACMV and hotspot ds-siRNA if their applied doses were less than 10 µg, although there were slight differences in disease severity (Figure 4B–D). We did not notice any influence of 2.5 and 5 µg hotspot ds-siRNA on viral infection, plants exhibited typical mosaic, distorting and stunting symptoms that were associated with ACMV infection. On the other hand, 10 µg hotspot ds-siRNA-treated plants were still susceptible to ACMV but showed milder systemic symptoms, often without stunting (Figure 4E). Remarkably, once the amount of hotspot ds-siRNA increased to 20 µg, those treated plants were almost immune to ACMV infection and remained asymptomatic throughout the course of the experiments (Figure 4F). Consistently, accumulation of viral DNA-A and DNA-B components was not detected in plants co-inoculated with ACMV and 20 µg hotspot ds-siRNA, but readily detectible and showed no obvious differences in plants infected with ACMV alone, or co-treated with less than 20 µg of hotspot ds-siRNA (Figure 4G–I). These data clearly demonstrated that a threshold level of hotspot ds-siRNA is required to have this dramatic effect on viral infection, a phenomenon that is associated with post-transcriptional RNA silencing-mediated innate antiviral defence [18].

### 3.4. Involvement of RDR6 in Hotspot ds-siRNA-Mediated Antiviral Defence

Plants have multiple *RDR* genes functioning in different biological processes. In *Arabidopsis*, there are six *RDRs* (*RDR1-6*), of which *RDR6* encodes the major RDR (RDR6) involved in antiviral RNAi [19,20,21]. *RDR6*-mediated RNAi forms an important defence against viral infection. It defends both differentiated and apical plant tissues from viral invasion, and silencing of *RDR6* causes plants to be hypersusceptible to virus infection [22,23,24]. Small RNA deep sequencing reveals that RDR6 is involved in vsiRNA biogenesis [6]. Recently, we have demonstrated that short-range intercellular trafficking of antiviral RNAi from a single epidermal cell to adjacent palisade and spongy parenchyma cells requires a functional *RDR6* gene [25]. Thus, it is possible that *RDR6* may be involved in the hotspot ds-siRNA-mediated viral resistance. To test this idea, we inoculated both wild-type *N. benthamiana* and NbRDR6i transgenic plants in which *RDR6* mRNA expression was reduced by RNAi knockdown [22,25,26] with 1.0 µg ACMV and 20 µg hotspot ds-siRNA. We found that the *RDR6*-deficient NbRDR6i plants, like the wild-type *N. benthamiana* plants, were always susceptible to viral infection when inoculated with ACMV alone, exhibiting a similar degree of symptom severity. Consistent with that we had previously observed (Figure 3; Table 2), when ACMV was co-inoculated with the hotspot ds-siRNA, the wild-type plants always showed resistance to viral infection, except in a few cases where one or two local chlorotic lesions were observed on inoculated leaves with these plants developing very mild curly systemic young leaves at a very late stage of infection at 12 weeks post-inoculation (Table 3). However, the hotspot ds-siRNA exhibited a much lower level of viral protection in NbRDR6i plants as ACMV was still able to infect the hotspot ds-siRNA co-inoculated NbRDR6i plants and systemic viral symptoms appeared in these plants at 10–14 dpi.

Consistent with the observed development of local and systemic symptoms, a similar amount of viral DNA accumulated in both wild-type and line NbRDR6i plants that were infected with ACMV. However, no accumulation of DNA-A (Figure 5A,B) and DNA-B (Figure 5C,D) was detected in young leaves of plants co-inoculated with ACMV and the hotspot ds-siRNA, but was readily detectible in those NbRDR6i plants co-inoculated with ACMV plus hotspot ds-siRNA. Interestingly, the level of viral DNA was less abundant in NbRDR6i plants that were co-inoculated with ACMV and hotspot ds-siRNA than with ACMV alone. Consistent with this, viral mRNA could be detected in wild-type *N. benthamiana* plants infected with ACMV alone, but not in plants where the co-inoculated hotspot vsiRNA suppressed viral infection. In contrast, viral mRNA was readily detectible in *RDR6*-deficient NbRDR6i plants that were infected with ACMV with or without the hotspot ds-siRNA treatment (Figure 6).

Taken together, these data revealed that knockdown of *RDR6* in NbRDR6i plants compromised the capability of hotspot ds-siRNA to combat virus infection, suggesting a direct or indirect requirement of a functional *RDR6* in the hotspot ds-siRNA-mediated antiviral response.

## 4. Discussion

Ds-siRNAs were first shown to mediate RNAi in *Caenorhabditis elegans* [27] and subsequently demonstrated in cultured mammalian cells [28]. Since then, the potential uses of ds-siRNAs in therapeutics and functional genomics across kingdoms have attracted great interest [29]. However, designing an effective siRNA is often challenging. Deep-sequencing technology has revealed siRNA hotspots, which are naturally susceptible to RNAi for siRNA biogenesis within endogenous cellular RNAs and viral genomes. This sequence information provides a useful guideline for designing and synthesising “native” hotspot siRNAs that could be active in triggering specific RNAi for virus resistance.

In this report, we tested such an idea. Through profiling vsiRNAs of the DNA geminivirus, ACMV, a devastating pathogen to cassava production, we identified vsiRNA hotspots from both viral genome components. There were five vsiRNA hotspots, three of which corresponded to the sense-strand and one to the complementary sense-strand of ACMV DNA-A, whilst only one obvious vsiRNA hotspot was found to be associated with the sense-strand of ACMV DNA-B (Figure 2). Using this information, we generated vsiRNA and provided compelling evidence that direct targeting one of the five hotspots by exogenous application of the synthetic vsiRNA confers plant resistance to virus infection (Figure 3; Table 2). It would be expected that the other hotspot vsiRNAs could have a similar impact on viral protection. Nevertheless, the particular vsiRNA hotspot, localising at the nucleotides 1458–1440 of the complementary-sense strand of ACMV DNA-A, seemed to be the hottest region from which vsiRNA could be generated. The hotspot vsiRNA in the double-stranded format, but not in single-stranded forms, effectively mediated antiviral defence (Figure 3; Table 2). This phenomenon is likely due to the fact that only the guide-strand of the vsiRNA duplex can be effectively incorporated into the RISC [4], which then targets and leads to cleavage of an RNA transcript coding for the four viral proteins (Rep, TrAP, REn and AC4) that are known to be essential for viral replication and infection [9,12,13].

A dose-threshold of the hotspot vsiRNA was required to be reached in order to mediate an effective antiviral defence (Figure 4), suggesting that the synthetic vsiRNA is able to activate the intracellular RNAi pathway to trigger primary RNAi. However, such a primary antiviral response may not be sufficient to withhold virus attack. Instead, secondary (and subsequent) RNAi needs to be activated through the action of RDR6 (Figure 7). This model is plausible and consistent with the finding that a functional *RDR6* is required for a robust resistance to ACMV infection (Figure 4, Figure 5 and Figure 6; Table 3). Moreover, RNA silencing is a plant innate defence mechanism and *RDR6* is an important component for the RNA-silencing machinery. Virus infection will trigger RNA silencing that leads to repression of viral mRNA in wild-type plants (e.g., Nb in this work). Plants defective in RNA silencing machinery such as *RDR6*-knockdown by RNAi will reduce their capacity to defend against virus infection. Consequently, viral gene expression (viral mRNA) and viral replication would increase in RDR6i plants. Thus, that the intensity of ACMV mRNA band in Nb extremely lower than that in RDR6i (Figure 6) would suggest that *RDR6*- and vsiRNA-dependent ACMV repression occurs even in susceptible tobacco plants.

## 5. Conclusions

In summary, we present compelling evidence that exogenous hotspot vsiRNAs can be used to control ACMV infection in plants. This is consistent with the role of siRNA in forming a dual defensive frontline for intra- and intercellular silencing to double-protect cells from virus infection in plants [25,30,31,32,33]. The scope of this work is very specific. However, it is also worth noting that hotspot vs. coldspot vsiRNAs have been found in other DNA and RNA viruses. In this regard, through a similar experimental design, we found that hotspot vsiRNAs were able to inhibit plant infection by Tomato mosaic virus, a positive sense ssRNA virus (Mohammed A.M. PhD thesis, University of Warwick 2012). Together, our data suggests that hotspot vsiRNAs may be deployed as a protection agent against viral infection. Moreover, vsiRNA hotspots may be good targets for engineering more effective ta-siRNA/miRNA-mediated resistance. Therefore, they could offer an alternative strategy to traditional breeding and contemporary transgenic resistance to protect plants and other hosts such as animals and even humans from destructive diseases caused by viruses and perhaps other pathogens including bacteria and fungi [34,35].

## 6. Patents

The data presented in this manuscript has been filed in a patent application: Hong Y. and Mohamed A.M. (2012) Control of viral pathogens using topical application of RNAi targeted at silencing hotspots in viral genome. UK Patent Ref. GB1011764.6.

## Figures and Tables

**Figure 1 biology-11-00714-f001:**
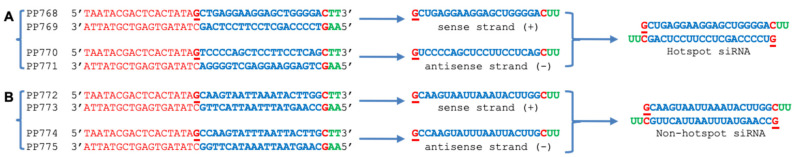
Design of ACMV siRNA. (**A**) hotspot siRNA. (**B**) Coldspot siRNA. Hotspot siRNA sequences in ACMV DNA-A start from 1440 to 1458 nt whilst coldspot siRNA sequences in ACMV DNA-A start from 1341 to 1359 nt on the viral complementary strand of ACMV DNA-A. T7 promoter-bearing oligonucleotides were annealed to form double-stranded templates for in vitro transcription to synthesise sense (+) and antisense (−) strands of small RNAs. The (+) and (−) RNA strands were complemented to produce siRNAs. T7 promoter is indicated red, siRNA blue and 3′-UU overhang green. The transcription start nucleotide is underlined. Notably, genome regions in which no or few vsiRNA is detectible are defined as “coldspot”. On the other hand, regions to which a large number of vsiRNAs can be mapped are regarded as “hotspot”. Hotspot and coldspot vsiRNAs were identified by bioinformatics analyses of the “454” sequencing data sets.

**Figure 2 biology-11-00714-f002:**
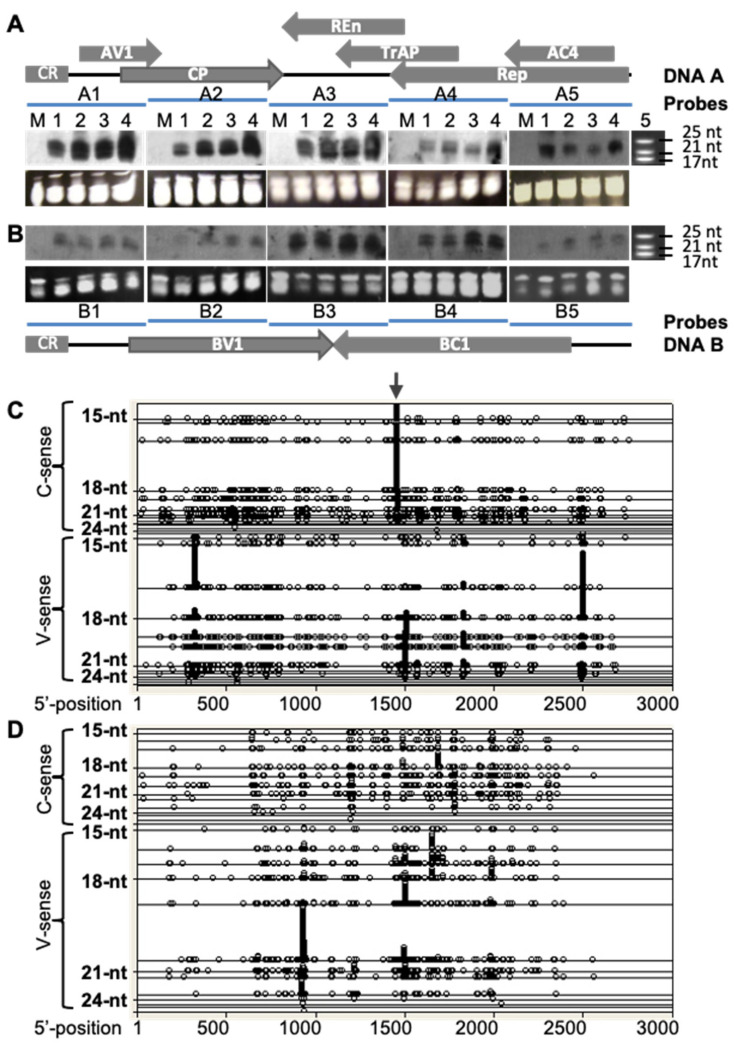
Profiling populations of vsiRNAs from ACMV (DNA-A and B)-infected plants. (**A**,**B**) Detection of ACMV vsiRNA by Northern blot. Total small RNAs were extracted from leaf tissues of mock-inoculated *N. benthaniana* (M), or plants infected with ACMV at 12 (lane 1), 15 (lane 2), 18 (lane 3) and 21 (lane 4) days post-inoculation. vsiRNAs of DNA-A (**A**) and DNA-B (**B**) were detected by Northern blots using probes hybridizing to separate regions of bipartite viral genome components. Genome organisations of DNA A and DNA B are indicated. Each line represents a specific probe (A1–5 and B1–5) that corresponds to approx. 560 bp fragment of the respective genome positioned below (**A**) or above (**B**). The sizes and positions of the microRNA markers (lane 5) are indicated. Loading controls are shown by GelRed-stained of total small RNAs in PAGE/8 M Urea gels below each of the small RNA blots. (**C**,**D**) Distribution of vsiRNAs within the ACMV genome. The profiles of vsiRNA corresponding to both virion (V)- and complementary (C)-sense strands of ACMV DNA A (**C**) and DNA B (**D**) were obtained by “454 deep-sequencing”. Each symbol represents 59 or 26 observations of siRNAs for DNA A (**C**) or DNA B (**D**), respectively. The length of vsiRNAs ranges from 15 to 29 nucleotides (nt). Distribution of each size of vsiRNAs is shown on a separate horizontal line. These blots show not only the vsiRNA distribution but also the size of vsiRNAs across ACMV genome sequences. Nucleotides of the ACMV genome are numbered as described [8]. The vsiRNA hotspot starting at nucleotide 1440 on the complementary sense strand of DNA A is indicated (arrow). Notably, vsiRNAs are likely derived from dsRNA precursors converted from viral mRNA.

**Figure 3 biology-11-00714-f003:**
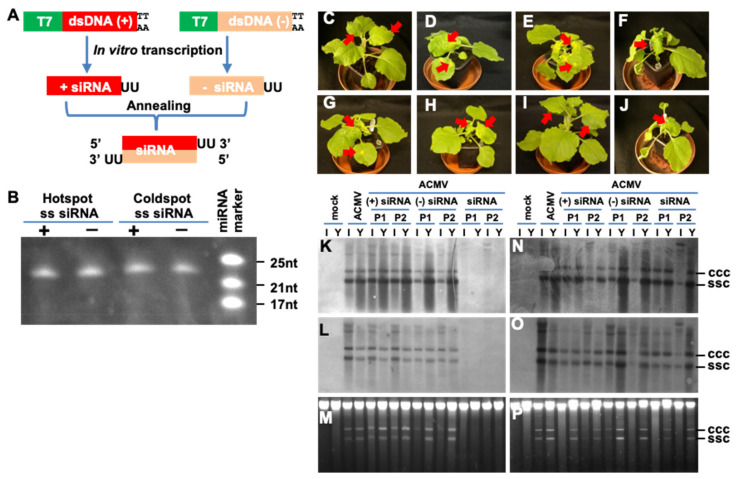
Hotspot siRNA inhibits ACMV infection. (**A**) Production of siRNA. Experimental strategy for in vitro synthesis of siRNA is outlined. Sense (+) and antisense (−) strand of siRNA with 3′-UU overhang were produced by in vitro transcription using T7 promoter-controlled dsDNA (+) and (−) templates. The (+) and (−) single-stranded (ss) siRNAs were mixed as 1:1 ratio and annealed to form siRNA. (**B**) Denaturing PAGE analyses of hotspot and coldspot siRNA. The positions and sizes of the miRNA markers are indicated. The transcribed siRNAs generated by in vitro transcription were treated with DNase, purified with Phenol/Chloroform extraction, and quantified before being checked on 15%PAGE/8M Urea gel electrophoresis. The sizes of these sRNAs are the same as predicted. (**C**–**J**) Impact of hotspot siRNA on ACMV infection. *N. benthamiana* plants were mock-inoculated (**C**), inoculated with ACMV (500 ng pCLV1.3A and 500 ng pCLV2B) (**D**) alone or co-inoculated with ACMV and 20 µg sense (+) strand of hotspot (**E**) or non-hotspot (**F**), antisense (−) strand of hotspot (**G**) or non-hotspot (**H**), or double-stranded (ds) hotspot (**I**) or non-hotspot (**J**) siRNA. Plants were photographed at 14 dpi. Typical ACMV symptoms including local chlorotic/yellow lesions, chlorosis and curling on systemic young leaves are indicated by arrows. Systemic healthy leaves in mock (**C**) and hotspot siRNA-protected (**I**) plants are also indicated. Hotspot ds-siRNA prevented plants from ACMV infection and plants remained symptomless at 3 months post co-inoculation. (**K**–**P**) Impact of hotspot siRNA on ACMV DNA replication. Total DNAs were extracted from systemic young (Y) and inoculated (I) leaf tissues of *N. benthaniana* plants with mock inoculation and of plants inoculated with ACMV alone, or together with sense (+), antisense (−), or siRNA corresponding to hotspot (**K**–**M**) or non-hotspot (**N**–**P**) vsiRNAs at 14 days post-inoculation. Duplicate plants (P1 and P2) were sampled for detection. Viral DNA was detected by probes specific to DNA-A (**K**,**N**) or DNA-B (**L**,**O**). GelRed staining shows equal loading of total DNA (**M**,**P**). The positions of single-stranded circular (ssc) and covalently closed circular (ccc) DNA are indicated.

**Figure 4 biology-11-00714-f004:**
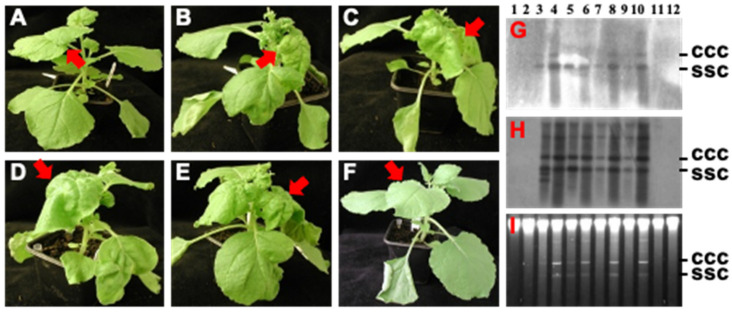
Threshold effect of hotspot siRNA on ACMV infection. (**A**–**F**) Plant infections. *N. benthamiana* plants were mock-inoculated (**A**), or inoculated with ACMV [500 ng of pCLV1.3A (DNA-A) and 500 ng of pCLV2B (DNA-B)] plus 0 (**B**), 2.5 (**C**), 5.0 (**D**), 10.0 (**E**) and 20.0 (**F**) µg hotspot siRNA. Plants were photographed at 14 dpi. Typical ACMV symptoms including chlorosis and curling on systemic young leaves are indicated by arrows. Systemic healthy leaves in mock (**A**) and hotspot siRNA-protected (**F**) plants are also indicated. (**G**–**I**) Southern detection of viral DNA accumulation. DNAs were extracted from inoculated leaves (**G**–**I**, lanes 1, 3, 5, 7, 9 and 11) and systemic young tissues (**G**–**I**, lanes 2, 4, 6, 8, 10 and 12) of plants with mock inoculation (lanes 1 and 2) and of plants inoculated with or inoculated with ACMV plus 0 (lanes 3 and 4), 2.5 (lanes 5 and 6), 5.0 (lanes 7 and 8), 10.0 (lanes 9 and 10) and 20.0 (lanes 11 and 12) µg hotspot siRNA. Viral DNA was detected by probes specific to DNA A (**G**) or DNA B (**H**). GelRed staining shows equal loading of total DNA (**I**). The positions of single-stranded circular (ssc) and covalently closed circular (ccc) DNA are indicated.

**Figure 5 biology-11-00714-f005:**
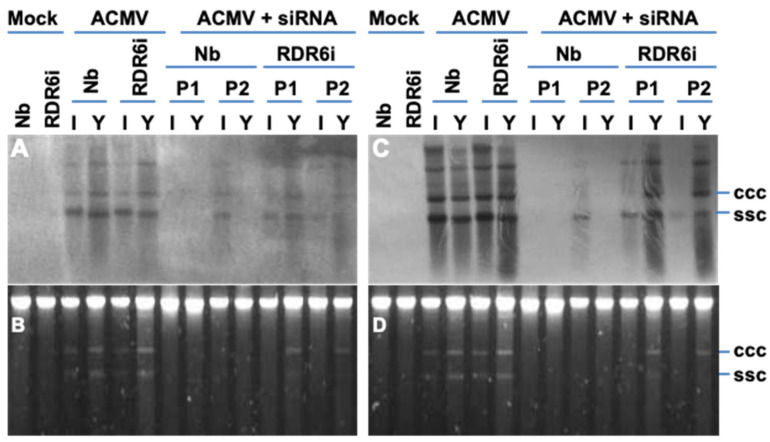
Involvement of *RDR6* in hotspot ds-siRNA-mediated suppression on ACMV DNA replication. (**A**–**D**) Southern blot detection and ACMV DNA accumulation. Total DNAs were extracted from systemic young (Y) and inoculated (I) leaf tissues of wild type *N. benthaniana* (Nb) and transgenic NbRDR6i (RDR6i) line plants with mock inoculation and of plants inoculated with ACMV alone (ACMV), or together hotspot ds-siRNA (ACMV + siRNA) at 14 days post-inoculation. Duplicate plants (P1 and P2) were sampled for detection. Viral DNA was detected by probes specific to DNA-A (**A**) or DNA-B (**C**). GelRed staining shows equal loading of total DNA (**B**,**D**). The positions of single-stranded circular (ssc) and covalently closed circular (ccc) DNA are indicated. It should be noted that the profiles of both DNA A and DNA B are almost identical. The different intensities between panel A (DNA-A) and panel C (DNA-B) should not be compared directly because the two blots were probed with different probes. It is possible that the different signal intensities may be due to various strength of the DIG-labelled probes. On the other hand, it is also worthwhile mentioning that ACMV DNA-A encodes essential function for viral DNA replication, and replication of DNA-B is completely dependent on DNA-A, although DNA-B is required for systemic spread of ACMV in plants. Therefore, the differential blot signals do not imply that ACMV lacking DNA-A can frequently be involved in systemic infection to young leaves since ACMV lacking DNA-A will not be able to establish any local or systemic infection.

**Figure 6 biology-11-00714-f006:**
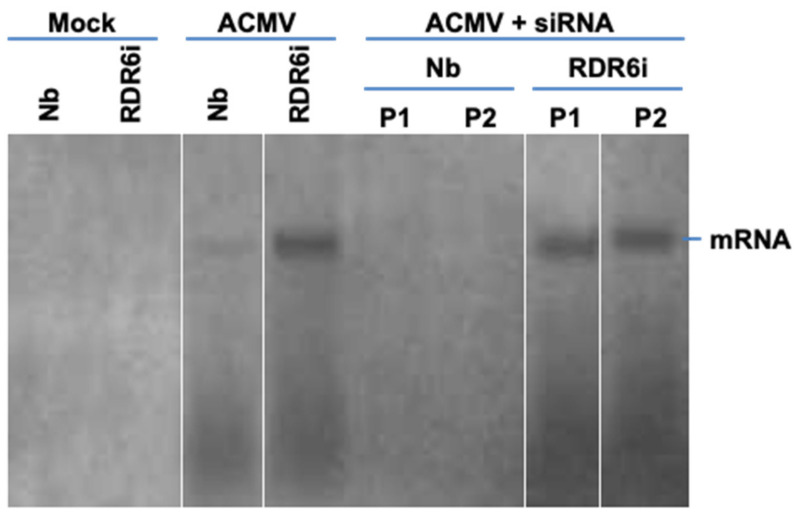
Northern blot detection of ACMV mRNA. RNA was extracted from leaf tissues of wild-type *N. benthaniana* (Nb) and transgenic NbRDR6i (RDR6i) line plants with mock inoculation and of plants inoculated with ACMV alone (ACMV), or together with hotspot ds-siRNA (ACMV + siRNA) at 14 days post-inoculation. Duplicate plants (P1 and P2) were sampled for detection. Viral mRNA was detected by probe A3 specific to DNA-A (Figure 2A). The position of viral mRNA is indicated. We would like to point out that this figure was re-assembled with different lanes separated by white lines from one same blot.

**Figure 7 biology-11-00714-f007:**
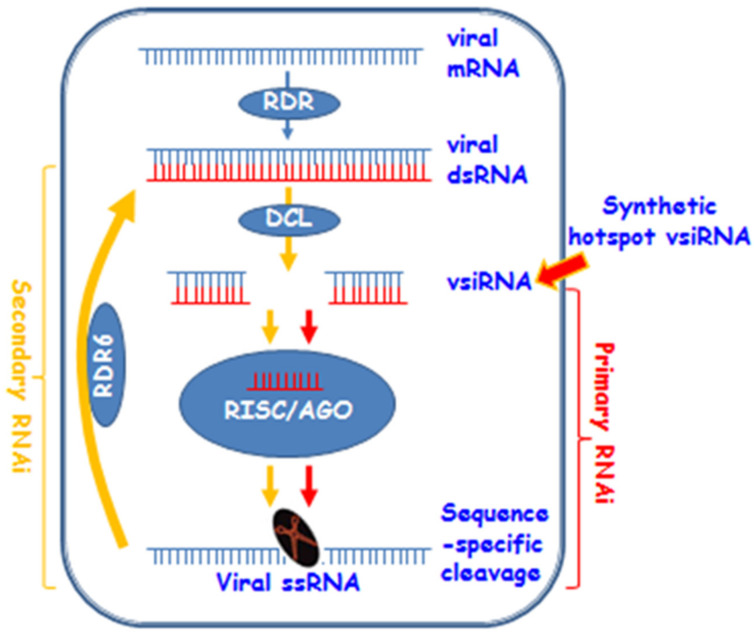
A model for synthetic hotspot vsiRNA-mediated antiviral RNAi. Viral RNA (mRNA) can be converted into dsRNA by plant RDRs such as RDR6. Then dsRNAs are diced into 21–24 nt vsiRNA by DCLs. At this stage, synthetic hotspot vsiRNA could enter the RNAi pathway. Subsequently, the guide strand of vsiRNA and AGOs are incorporated into the RISC. The RICS targets the viral ssRNA (or mRNA transcribed from viral genome) for sequence-specific cleavage to cause the primary RNAi effect. Primary RNAi can be amplified via RDR6 to trigger the secondary RNAi to increase viral defence in plants. It should be noted that this model does not exclude other possible mechanisms underpinning how vsiRNA might confer plant resistance against virus infection. For instance, vsiRNA could block virus entry to or inhibit viral genome replication in host cells.

**Table 1 biology-11-00714-t001:** Primers used in this study.

Primers	Sequences	Application
PP684	5′TTGGAGACACTCAACTAGAGACACTCTTGAGCAT3′	Probe A1
PP685	5′TGACGTGGACAGTGGGGGCAGTAGCACGGTTCCTG3′	Probe A1
PP686	5′ACTGTCCACGTCACAAATCGAAAACGG3′	Probe A2
PP687	5′TCATACTTCCCTGCCTCCTGATGATTGTATGTCAC3′	Probe A2
PP688	5′GGCAGGGAAGTATGAGAATCACACAGAGAATG3′	Probe A3
PP689	5′GGATGTTCATATTACCTCCATATCAACTGCTTCAA3′	Probe A3
PP690	5′GTAATATGAACATCCACAGACAAGATCCACTCT3′	Probe A4
PP691	5′TTCCCATGTTCTTCCTTTGACCAAGTTCCTGTTGA3′	Probe A4
PP692	5′TCAAAGGAAGAACATGGGAAGGGAGAAACATAAG3′	Probe A5
PP693	5′TCTAGTTGAGTGTCTCCAATTGACTTGGTCAACATG3′	Probe A5
PP758	5′TAGAGAGAGACGCTCTCAACTGGAGACACACTTG3′	Probe B1
PP759	5′TGTATTTCGCACCCTATATATAATACATAA3′	Probe B1
PP760	5′ATACAGGAGTTGGAGAATATCATTTATTGGAAGTA3′	Probe B2
PP761	5′TATTCAATAATCTTAAATTAACGTAACAAGCGGAA3′	Probe B2
PP762	5′GAATAATCAACAGCATCGATATAGGGTATT3′	Probe B3
PP763	5′CCACAATTGGGCGCTATACAAGCATGAGAT3′	Probe B3
PP764	5′TGTGGATTTTGTAGCCCATGTTTCTCCTGGTT3′	Probe B4
PP765	5′TTCCTTCGACCCTTGAGAGAACAAGGGTACGTAT3′	Probe B4
PP766	5′AGGAAATTGAAGTGTAATCGGCGATTCATC3′	Probe B5
PP767	5′TCCAGTTGAGAGCGTCTCTCTCTAACTTTCTCTC3′	Probe B5
PP768	5′TAATACGACTCACTATAGCTGAGGAAGGAGCTGGGGACTT3′	Hotspot (+) strand siRNA
PP769	3′ATTATGCTGAGTGATATCGACTCCTTCCTCGACCCCTGAA5′	Hotspot (+) strand siRNA
PP770	5′TAATACGACTCACTATAGTCCCCAGCTCCTTCCTCAGCTT3′	Hotspot (−) strand siRNA
PP771	3′ATTATGCTGAGTGATATCAGGGGTCGAGGAAGGAGTCGAA5′	Hotspot (−) strand siRNA
PP772	5′TAATACGACTCACTATAGCAAGTAATTAAATACTTGGCTT3′	Coldspot (+) strand siRNA
PP773	3′ATTATGCTGAGTGATATCGTTCATTAATTTATGAACCGAA5′	Coldspot (+) strand siRNA
PP774	5′TAATACGACTCACTATAGCCAAGTATTTAATTACTTGCTT3′	Coldspot (−) strand siRNA
PP775	3′ATTATGCTGAGTGATATCGGTTCATAAATTAATGAACGAA5′	Coldspot (−) strand siRNA

**Table 2 biology-11-00714-t002:** Suppression of viral infection by hotspot ds-siRNA.

Experiment	I		II	
	Hotspot	Coldspot	Hotspot	Coldspot
Mock	0/2	0/2	0/3	0/3
ACMV	2/2	2/2	3/3	3/3
ACMV/(+) strand	2/2	2/2 *	3/3	3/3
ACMV/(−) strand	2/2	2/2	3/3	3/3
ACMV/siRNA	0/2	2/2	0/4	3/3

* One plant showed mild curly, chlorosis and mosaic symptoms in systemic leaves. The number of plants exhibiting symptoms out of the number of plants infected with ACMV +/− hotspot or coldspot, ss- or ds-siRNA in different experiments.

**Table 3 biology-11-00714-t003:** Hotspot siRNA-mediated antiviral protection requires *RDR6*.

Experiment		I	II	III
Mock	Nb	0/2	0/2	0/4
	NbRDR6i	0/2	0/2	0/4
ACMV	Nb	3/3	4/4	4/4
	NbRDR6i	3/3	4/4	4/4
ACMV/ siRNA	Nb	0/3 *	0/4 *	0/3 *
	NbRDR6i	3/3	4/4	3/3

* One plant showed one or two local lesions on inoculated leaves and developed subsequently mild curly in systemic leaves. The number of plants exhibiting symptoms out of the number of plants infected with ACMV +/− hotspot ds-siRNA in different experiments.

## Data Availability

Not applicable.

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
