# Peer review of "Hotspot siRNA Confers Plant Resistance against Viral Infection"

_biology, 2022, doi:10.3390/biology11050714_

Round 1
Reviewer 1 Report
This manuscript examined that a short dsRNA synthesized with siRNA-hotspot region of ACMV DNA-A impacts on repression of ACMV proliferation via RDR6-mediated gene silencing in tobacco plants.
First, the authors performed high-throughput sequencing of small RNAs extracted from ACMV-infected tobacco leaves. They detected two main sizes of vsiRNAs in 21 and 24-nt lengths, of which 24nt-vsiRNA derived from the middle of the DNA-A antisense strand complementary to virion strand was prominent (hotspot (HS)-vsiRNA). In contrast, most of DNA-A and -B regions except for vsiRNA hotspot were silent in vsiRNA production (coldspot (CS)).
When ACMV was infected to tobacco leaves with 20 µg synthesized single-stranded (ss) HS-vsiRNAs or double-stranded (ds) HS-vsiRNA, only the dsHS-vsiRNAs repressed ACMV disease symptoms (Fig. 2). Consistently, ACMV DNAs were significantly reduced in both infected and systemic young leaves in the plant infected with 20µg dsHS-vsiRNAs (Fig. 3). A suppressive effect of dsHS-vsiRNAs in ACMV proliferation and symptom was injected dsHS-vsiRNA dose-dependent, and a critical threshold was observed between 20 and 10 µg (Fig. 4). RNAi-knockdown of RDR6 resulted in reduction of the suppressive effect of HS-vsiRNAs on ACMV local and systemic infection (Fig. 5) and of ACMV mRNA expression (Fig. 6). Then the authors proposed a model in which RDR6-mediated dsHS-siRNA from a viral genomic region reinforces cleavage of viral mRNAs, resulting in suppression of ACMV proliferation and systemic infection (Fig. 7).
I understand the theme of this manuscript is very important for plant pathology and breeding programs. However, I have several questions and found some suspicious things in the results.
(1) Fig. 3B,
The authors performed the in vitro transcription of sense and antisense ssHS-vsiRNAs, and explained it is T7 promoter-controlled in the legend of Fig. 3. However, to show it really T7-promoter-controlled, a negative control, for example, a reaction without T7 RNA polymerase, should be added.
(2) L84-86,
Describe more about the method for co-inoculation of ACMV and siRNAs together into tobacco leaves. Wezel et al. (2002) (ref.12) infected ACMV to tobacco leaves, but without siRNAs. Vanitharani et al. (2003) (ref.17) co-infected ACMV and siRNAs to protoplast, but not to leaves.
(3) Fig. 3K and L246-249,
I could only find faint smeared signals in ds-siRNA lanes I (inoculated leaves) in Fig.3K, but neither of CCC or SSC band indicating the residual of inoculum ACMV DNA. Thus it is difficult to agree with the authors' supposition, "simultaneously introducing the synthetic hotspot ds-siRNA activated the host RNAi defence mechanism that effectively prevented ACMV from establishing a systemic infection in plants". If little ACMV was detected in inoculated leaves, it is plausible that dsHS-siRNA infection represses ACMV proliferation in inoculated leaves, rather than systemic infection in young leaves.
(4) RDR6 RNAi plants,
There is no evidence showing RDR6 mRNA expression is actually reduced by RNAi knockdown.
(5) Fig. 5,
In RDR6i knockdown plants, I wonder why the amount of DNA-B in young leaves seems to apparently be more than that in inoculated leaves, while the amount of DNA-A seems similar between I and Y leaves. Is it possible that ACMV lacking DNA-A can frequently be involved in systemic infection to young leaves? As another possibility, I wonder if the authors adjust the DNA amount appropriately among all gel lanes or not. Judged from the signal intensity of host genome DNAs stuck at the top (panels B, D), it doesn't seem adjusted precisely among lanes. Anyway, more explanation about the above points should be required.
(6) Fig. 6,
At first, I found several artificial straight lines between several lanes, which might be a sign of inappropriate combination of different photos. Separate them appropriately for readers to be able to recognize the photos originate from different blotting experiments with each other.
Second, the Northern blot in this figure lacks a loading control to show the same amount of total RNAs was loaded in every lane.
Third, in the ACMV (w/o siRNA) lane, why is the intensity of ACMV mRNA band in Nb extremely lower than that in RDR6i? Does this mean that RDR6- and vsiRNA-dependent ACMV repression occurs even in susceptible tobacco plants? There is no description about this point in the text.
Reviewer 2 Report
in this article, authors identified vsiRNA hotspots in ACMV genome and found that coinfection of synthetic dsDNA hotspot siRNA protected the plants from ACMV infection. The article is not rigorous in methodology and lack substantial evidence to support their claim. More importantly, it looks that authors mixed several blots to make one figure (Figure 6), and most of time loading does not look same across all the samples.
Other comments;
Figure 1A-B there is no loading control.
Figure 3C-J- I don’t see the symptoms as image presented are not good (just showing plants). Authors should present the images that clearly support their results and claims. Same the images presented in figure 4 are not optimal, like from infected to protected??
Figure 3 M-P. it seems that loading is not the same across all the samples, and this was the problem with other blots in the manuscript
Authors presented the data with RDR6i but did not show any phenotype and they did not provide enough support their claims.
Reviewer 3 Report
In this work, Mohamed and colleagues explored the use of dsRNAs as a strategy to control virus infection defining targets based on siRNA-enrichment profile.
The paper is very interesting and well written. However, some questions should be addressed before its ready for publication.
Minor
What are about discuss in the introduction and discussion other DNA viruses in plants and other organisms... the "hotspot" profile is observed in all viruses or only DNA?
Coldspot means that the small RNAs are depleted in that region? There is other regions with "average" production of small RNAs... how are those called?
Why "454" strategy? What are the limitations of this sequencing machine?
Line 230 sounds odd.
How hotspot regions were determined? manually?
Figure 3 MUST be enlarged to allow better visualization of the symptoms. Its not clear n the current size.
Major
Figure 2, representation of the density of small RNAs is confuse. Please use lines or "density". Dots do not allow proper evaluation.
Where are the plots of small RNA distribution ? Barplot with sizes in the X axis and amount of that respective size in Y axis. Its mandatory for works analyzing small RNAs. Also its important to show 5' base preference to allow the analysis of possible origin of the small RNAs!
Profile of small RNAs (size distribution, base preference and density along the sequence) derived from ds-siRNA precursors?
Reviewer 4 Report
Virus infected plants usually showed up through virus derived small interfering RNA, which is as a result of complex fight between the plant and virus. This fight is termed RNAi, and has been an important tool for designing therapeutic approaches for virus related diseases in plants and animals. The authors, carefully designed and illustrated vsiRNA hotspots in the genome of ACMV (devastating to cassava production) in Africa. The approach used in the identification of vsiRNA hotspts was well designed, data analysed and presented accordingly. Their finding provides a promising alternative in the development program of resistant cultivar. In view of manuscript improvement, I recommend some minor changes as comments and suggestions below:
1- I suggest the authors to include a short explanation on how the small RNA sequenced data was analysed (including tools and programs).
2- Line 98: close the space between ...EDTA) and at
3- Line 100-101: consistency in the word hybridised/hybridized...reference to line182.
4- Line 123: The hanging UUs showed sky blue on printed version but not green, please check this out.
5- Line 168: Close the space between ...D) and Moreover,...
6- Lines 210-211: in vitro to in vitro
7- Line 281: 14dpi to 14 dpi
8- Line 346: (Nb)and to (Nb) and...
Reviewer 5 Report
In this manuscript, the authors show that direct targeting a hotspot by synthetic vsiRNA confers plant resistance to African cassava mosaic virus (ACMV), were synthesised. Further, they demonstrat that the hotspot vsiRNA-mediated virus resistance had a threshold effect and required an active RDR6. These data show that hotspot vsiRNAs bear a functional significance on antiviral RNAi. The data is interesting and convincing. The manuscript is well-written and can be considered for the acceptance.
Round 2
Reviewer 1 Report
The regions that I pointed out are largely improved. So the current version is worthy to be published. However, I still recommend in Fig. 6, the authors use the improved image of norther blot in which multiple photos were separated appropriately with white lines, as shown in the answer sheet. As a minor point, within newly added sentences in Fig. 6, "... between panel A (DNA-A) and panel B (DNA-B)..." is a mistake of "... between panel A (DNA-A) and panel C (DNA-B)...".Author Response
Please see the attachment.

Reviewer 2 Report
revisions made by authors are not satisfactory.
Reviewer 3 Report
Since the authors did not make any effort to address the major points raised by this reviewer nor provided a reasonable justificative, I am not comfortable with further revising the manuscript.
